# Genomic Comparison of Insect Gut Symbionts from Divergent *Burkholderia* Subclades

**DOI:** 10.3390/genes11070744

**Published:** 2020-07-03

**Authors:** Kazutaka Takeshita, Yoshitomo Kikuchi

**Affiliations:** 1Faculty of Bioresource Sciences, Akita Prefectural University, Akita City, Akita 010-0195, Japan; 2Bioproduction Research Institute, National Institute of Advanced Industrial Science and Technology (AIST), Hokkaido Center, Sapporo, Hokkaido 062-8517, Japan; y-kikuchi@aist.go.jp; 3Graduate School of Agriculture, Hokkaido University, Sapporo, Hokkaido 060-8589, Japan

**Keywords:** insect-microbe interaction, gut symbiosis, stink bugs, bordered plant bugs, *Burkholderia*, environmental acquisition, evolution, comparative genomics

## Abstract

Stink bugs of the superfamilies Coreoidea and Lygaeoidea establish gut symbioses with environmentally acquired bacteria of the genus *Burkholderia* sensu lato. In the genus *Burkholderia*, the stink bug-associated strains form a monophyletic clade, named stink bug-associated beneficial and environmental (SBE) clade (or *Caballeronia*). Recently, we revealed that members of the family Largidae of the superfamily Pyrrhocoroidea are associated with *Burkholderia* but not specifically with the SBE *Burkholderia*; largid bugs harbor symbionts that belong to a clade of plant-associated group of *Burkholderia*, called plant-associated beneficial and environmental (PBE) clade (or *Paraburkholderia*). To understand the genomic features of *Burkholderia* symbionts of stink bugs, we isolated two symbiotic *Burkholderia* strains from a bordered plant bug *Physopellta gutta* (Pyrrhocoroidea: Largidae) and determined their complete genomes. The genome sizes of the insect-associated PBE (iPBE) are 9.5 Mb and 11.2 Mb, both of which are larger than the genomes of the SBE *Burkholderia* symbionts. A whole-genome comparison between two iPBE symbionts and three SBE symbionts highlighted that all previously reported symbiosis factors are shared and that 282 genes are specifically conserved in the five stink bug symbionts, over one-third of which have unknown function. Among the symbiont-specific genes, about 40 genes formed a cluster in all five symbionts; this suggests a “symbiotic island” in the genome of stink bug-associated *Burkholderia*.

## 1. Introduction

Symbioses with microorganisms are ubiquitously found in various animals and plants in nature. Insects are the most diverse animal group in the terrestrial ecosystems and consist of over one million described species, of which almost half are associated with mutualistic microorganisms [1,2,3,4]. These microbial symbionts play important metabolic roles and provide adaptive traits such as heat tolerance and pathogen resistance that have allowed their host insects to expand habitats and become diverse [5,6,7]. Although knowledge of the biological importance of symbionts has increased, information on how beneficial symbionts evolved from environmental microorganisms are still scarce. Most characterized insect symbionts are vertically transmitted from mother to offspring and have highly adapted to host in vivo environments, which is often coupled with symbiont genome reduction [8,9]. The longer evolutionary history of most of these characterized symbiotic associations has made it difficult to reveal the evolutionary transition from non-symbionts to symbionts [9]. There are few insect systems suitable for addressing this evolutionary issue, in which both the symbionts and their closely related free-living strains are available as in vitro cultures with corresponding whole-genome sequence data.

Several phytophagous stink bugs have developed crypt-bearing symbiotic organs in their midgut that densely harbor gut symbiotic bacteria [3,10,11,12,13]. The bean bug *Riptortus pedestris* (superfamily Coreoidea; family Alydidae), which is the most intensively and comprehensively investigated species among stink bugs, hosts a culturable symbiont, *Burkholderia insecticola* (class Betaproteobacteria) in the lumen of their midgut crypts [11,14,15]. Unlike many other insects that vertically transmit their symbionts, the *Burkholderia* symbiont of the bean bug is not essential to their hosts and is not vertically transmitted. Instead, hatchlings of the bean bug are initially free from *Burkholderia* symbionts that are acquired later in life from the ambient soil environment [16]. Sorting the appropriate partner from enormously diverse soil microbiota is achieved in two steps: (i) physical selection at the narrow entrance of the crypt-bearing symbiotic organ, called the constricted region, and (ii) a subsequent microbial competition within the lumen of the symbiotic organ [17,18,19,20]. The *Burkholderia* symbiont has beneficial effects on the host’s fitness, although it is not essential. For example, infected stink bugs grow larger and lay a higher number of eggs than non-infected ones [16,18]. Moreover, some *Burkholderia* symbiont strains can degrade substances including insecticides; such insecticide-degrading *Burkholderia* symbiont thus confers insecticide resistance to their hosts [21,22].

The genus *Burkholderia* sensu lato consists of over 100 species and had been phylogenetically grouped into three main clades. Briefly, the three clades were as follows: the “*Burkholderia cepacia* complex and *Burkholderia pseudomallei* (BCC&P)” clade included many animal and plant pathogenic species; the “plant-associated beneficial and environmental (PBE)” clade included environmental species, plant growth-promoting rhizobacteria, and leguminous nodule symbionts; and the “stink bug-associated beneficial and environmental (SBE)” clade included environmental species and midgut symbionts of various stink bugs [11,23,24,25]. The symbiont of the bean bug, *B. insecticola*, belongs to the SBE clade. As the efforts for better taxonomic classification of this diverse bacterial genus have been continued, the genus *Burkholderia* has been subdivided into six nominated genera based on their phylogenetic relationships [12,26,27,28,29,30]. The BCC&P, PBE, and SBE clades currently correspond to *Burkholderia* sensu stricto, *Paraburkholderia*, and *Caballeronia*, respectively. The remaining three genera are *Robbsia*, *Mycetohabitans*, and *Trinickia*. Considering the ongoing taxonomic reclassification, in this manuscript we refer to the six subdivided genera as the genus *Burkholderia*. Given the close phylogenetic relationships between symbiotic and non-symbiotic environmental species and the unique, free-living life stage of the SBE *Burkholderia*, this stink bug-*Burkholderia* symbiosis provides a valuable opportunity for tracking the evolutionary transition of a free-living environmental bacterium into an insect gut symbiont.

Previous studies, including our own, have revealed that the stink bug-*Burkholderia* symbiosis with environmental symbiont acquisition originated at the common ancestor of the stink bug superfamilies Coreoidea, Lygaeoidea, and Pyrrhocoroidea [31,32,33,34]. However, the evolution of this interesting symbiotic system was found to be more complicated than we expected before; recent studies have shown that the symbiotic *Burkholderia* in the family Largidae of the Pyrrhocoroidea is phylogenetically distinct from that of the other two superfamilies [31,33]. While the symbionts of the Coreoidea and Lygaeoidea are members of the SBE clade, those of largid bugs are members of the PBE clade, wherein the largid bug-associated *Burkholderia*, as well as closely related species isolated from rhizosphere, form a subclade named the “insect-associated PBE (iPBE)” clade [31].

Two strains of the SBE *Burkholderia* have been isolated from the bean bug *R. pedestris* and one SBE strain from the seed bug *Togo hemipterus* (superfamily Lygaeoidea; family Rhyparochromidae); we also have sequenced their complete genomes [35,36,37]. However, available isolated strains of the largid bug-associated iPBE *Burkholderia* and their genomic information were lacking due to difficulties in culture by the simple plating of gut contents of the symbiotic organs. In this study, we succeeded in isolating iPBE *Burkholderia* symbionts from the midgut crypts of the bordered plant bug *Physopelta gutta* (superfamily Pyrrhocoroidea; family Largidae; Figure 1A) by applying the organ culture method [38]; we then determined the complete genome sequences of the two isolates. Furthermore, we performed a comparative analysis of the newly sequenced iPBE symbiont genomes with those of other *Burkholderia* species including the SBE *Burkholderia* symbionts, highlighting the genes and functions commonly conserved among stink bug-associated iPBE and SBE *Burkholderia* symbionts.

## 2. Materials and Methods

### 2.1. Insect Samples

Samplings of *P. gutta* were conducted in Okinawa prefecture, Japan in June 2016 and July 2019 (Table 1). The insects were fed with sunflower and peanut seeds and distilled water containing 0.05% ascorbic acid until examination. *Burkholderia* symbionts were isolated from the gut of two adult females in 2016, and of three adult females and three adult males in 2019.

### 2.2. Isolation of Symbiotic Bacteria from the Midgut Crypts

Wild-captured adult insects were dissected in phosphate-buffered saline (PBS: 137 mM NaCl, 2.7 mM KCl, 8.1 mM Na_2_HPO_4_, 1.5 mM KH_2_PO_4_ [pH 7.4]) with micro-scissors and micro-tweezers under a dissection microscope (S9 D, Leica Microsystems, Wetzlar, Germany); whole midguts were photographed with a digital microscope camera (MC170 HD, Leica Microsystems). To isolate *Burkholderia* symbionts from the midgut of the bordered plant bug, the organ culture method originally developed by Xu et al. [38] was employed with a simple modification. The crypt-bearing midgut 4th section (Figure 1B), where the symbionts are specifically harbored, was isolated and transferred to a tissue-culture test plate (24-well; TPP Techno Plastic Products AG, Trasadingen, Switzerland). After washing three times with PBS and two times with yeast-glucose (YG) medium (0.5% yeast extract, 0.4% glucose, 0.1% NaCl), the symbiotic organ was pre-cultured in YG medium instead of Grace’s insect cell culture medium, which was applied in the original method [38], at 27 °C for 2–3 days. Subsequently, the pre-cultured symbionts were retrieved from the symbiotic organ by pipetting, streaked on YG-agar plate, and incubated at 27 °C for 2 days. After taxonomic identification by sequencing the 16S rRNA gene, new isolates of the *Burkholderia* symbionts were cultured overnight in YG medium at 28 °C and stocks of them were stored in 20% glycerol at −80 °C.

### 2.3. Sequencing of Bacterial 16S rRNA Gene

At least six isolates for each insect sample were randomly selected and subjected to diagnostic PCR for the specific detection of the PBE *Burkholderia* as described in our previous study [31]. Then, positive isolates were subjected to direct PCR with TaKaRa Ex Taq Hot Start Version (TaKaRa Bio, Kusatsu, Japan) for taxonomic identification by sequencing the 16S rRNA gene. All the primers used in this study are listed in Appendix A. The PCR reaction mixture was prepared according to the manufacturer’s instructions. The PCR conditions in the order of initial denaturation, denaturation, annealing, elongation, and final elongation were as follows: 94 °C for 2 min; 35 cycles of 98 °C for 10 s, 52 °C for 30 s, and 72 °C for 2 min; and 72 °C for 3 min. PCR products were cleaned up with Exonuclease I (*Escherichia coli*) (New England Biolabs, Ipswich, MA, USA) and Alkaline Phosphatase (Shrimp) (TaKaRa Bio). Sanger sequencing was performed by Macrogen Japan (Kyoto, Japan). Assembly of the Sanger-sequenced reads was performed as previously described [31]. The 16S rRNA gene sequences of the isolates were then queried for homology with blastn of the BLAST+ 2.5.0 [40] against the NCBI 16S rRNA sequence database to confirm whether these isolates are of *Burkholderia* or not. To identify the closest *Burkholderia* species, the sequences were subsequently analyzed with EzBioCloud [39].

### 2.4. Genome Sequencing, Assembly, Finishing, and Annotation

The two isolates of the iPBE symbionts, PGU16 and PGU19, were subjected to whole-genome sequencing. Genomic DNA from an overnight culture in YG medium at 28 °C was extracted using the Wizard^®^ Genomic DNA Purification Kit (Promega, Madison, WI, USA) according to the manufacturer’s instructions. Preparation of a 20-kb library with DNA Template Prep Kit 3.0 (Pacific Biosciences of California, Menlo Park, CA, USA), PacBio RSII sequencing with the P6-C4 chemistry (Pacific Biosciences of California), and de novo assembly with HGAP v3.0 [41] were performed by Macrogen (Seoul, South Korea). To close a gap in one of the replicons (Chromosome 2) of PGU16, PCR amplification with KOD -Plus- Ver.2 (Toyobo, Osaka, Japan) and Sanger sequencing were performed with specifically designed primers (Appendix A). The PCR reaction mixture was prepared according to the manufacturer’s instructions, and the PCR conditions were as follows: 94 °C for 2 min; 35 cycles of 98 °C for 10 s, 60 °C for 10 s, and 68 °C for 1.5 min; and 68 °C for 3 min. The PCR products were purified with the Wizard^®^ SV Gel and PCR Clean-Up System (Promega) and then sequenced in both directions. The Sanger-sequenced reads were manually inspected and used for circularization. The assignment of each replicon of the circularized genomes to either chromosome or plasmid was based on the comparison at protein level with the genome of *Burkholderia caribensis* MWAP64 [42].

Finished genomes were annotated by using DFAST v1.2.4 [43] with default settings. The protein sequences from each genome were also queried for homology with blastp against the Clusters of Orthologous Groups of proteins (COGs) database with a threshold E-value of 10^−3^ and were functionally categorized [44]. Circular genomes were visualized with Circos v0.69-8 [45].

### 2.5. Phylogenetic Analyses

To determine the phylogenetic positions of the newly isolated *Burkholderia* symbionts, phylogenetic analyses with the maximum likelihood (ML) method were conducted based on the 16S rRNA gene and genome sequences. The 16S rRNA gene sequences of the isolates and uncultured *Burkholderia* derived from the midgut crypts of largid stink bugs [31,33], as well as those of representative *Burkholderia* species, were aligned by using SINA (the SILVA Incremental Aligner) v1.2.11 [46]. For genome-based phylogeny, genome sequences of the representatives *Burkholderia* were downloaded from GenBank (Appendix A). Ninety-two up-to-date bacterial core genes (UBCGs) were extracted from the data by using the UBCG pipeline [47]. After translation into protein sequences, 70 UBCGs that were found in all the analyzed *Burkholderia* (Appendix A) were aligned independently with L-INS-i of MAFFT v7.455 [48].

Gap-including and ambiguous sites were removed from both types of multiple alignments. Phylogenetic relationships were reconstructed with RAxML v8.2.12 [49]. The GTR + Γ model was applied for 16S rRNA gene phylogeny. The WAG + Γ model was applied for genome-based phylogeny and the parameters for each gene partition were calculated independently. The bootstrap values of 1000 replicates for all internal branches were calculated with a rapid bootstrapping algorithm [50].

### 2.6. Comparative Genomics

To calculate the values of pairwise digital DNA-DNA hybridization (dDDH) between the isolates and the iPBE *Burkholderia* species, i.e., *Burkholderia phymatum*, *B. caribensis*, *Burkholderia hospita*, *Burkholderia terrae* and *Burkholderia steynii*, the genome sequences were analyzed with GGDC 2.1 [51]. To perform comparative genomic analysis, protein sequences of the related *Burkholderia* genomes were downloaded from GenBank (Appendix A). The orthologous gene clustering of the five *Burkholderia* symbionts of stink bugs listed in Table 2, *Burkholderia phytofirmans* PsJN (PBE), and *B. pseudomallei* K96243 (BCC&P) were performed with OrthoFinder v2.3.11 [52] with the options “-S blast -M msa -T raxml”. The COG functional categories of the orthogroups were assigned based on the functional categories of genes of PGU19 and *B. insecticola* RPE64, which were included in the orthogroup. The genes involved in the virulence in *B. pseudomallei* K96243 [53,54], in symbiotic association with plants in *B. phytofirmans* PsJN [55,56], and in symbiosis with stink bugs in *B. insecticola* RPE64 [17,57,58,59,60,61,62,63,64,65,66] were searched in the resultant orthogroups and their orthologs were identified.

### 2.7. Data Availability

The nucleotide sequences of 16S rRNA gene determined in this study have been deposited in DDBJ/ENA/GenBank under the accession no. LC547415 to LC547419 (see Table 1). The genome sequences of PGU16 and PGU19 have been deposited in DDBJ/ENA/GenBank under the accession no. AP023174 to AP023178 and AP023179 to AP023183, respectively. The raw sequence reads have been deposited in the DDBJ Sequence Read Archive under the accession no. DRA010199.

## 3. Results

### 3.1. Isolation of Burkholderia Symbiont from the Midgut Crypts of a Largid bug

As reported in our previous study [31], the midgut of *P. gutta* consisted of four morphologically distinct sections, the midgut 1st section (M1) to the crypt-bearing midgut 4th section (M4), and that its *Burkholderia* symbionts are specifically localized inside tubular-type crypts of M4 (Figure 1B,C). Although the crypts were translucent before pre-culture, the M4 crypts became swollen and their color changed into cream after one- to three-day pre-culture (depending on the samples) in YG medium at 27 °C (Figure 1D,E). By streak-plating the content of the pre-cultured M4 crypts, bacterial colonies were successfully cultured on YG-agar plates. Among the eight pre-cultured crypt samples, colonies were observed from only five samples.

Isolated colonies from the five samples were identified as *Burkholderia* by diagnostic PCR and by querying their 16S rRNA gene sequences against nucleotide databases through blastn (Table 1). According to our previous study based on culture-independent approach, where symbiotic microbiota of largid species showed some level of diversity within the individuals [31], we expected that several genetically distinguishable strains would be isolated from a single insect sample. Therefore, at least six randomly selected colonies from each insect sample were sequenced. However, no sequence diversity between colonies derived from the same insect sample was observed. This loss of diversity might have been due to the pre-culture step. The described closely related species from the five isolates were *B. caribensis* or *Burkholderia sabiae*; this is consistent with our previous study based on culture-independent methods [31], although it remains unclear how much individuals in an insect population possess these isolates. The sequences of the isolates PGU16 and F2 were identical even though the host individuals were sampled with a 3-year interval (Table 1). The lowest sequence identity between the five isolates was 98.2% (between PGU19 and F1).

### 3.2. Phylogenetic Affiliation of the Isolated Symbionts

To determine the phylogenetic position of the isolated iPBE *Burkholderia* from *P. gutta*, a 16S rRNA gene-based phylogenetic analysis was performed. Figure 2 shows the ML tree of the five isolates from the largid bug and related *Burkholderia* species/clones based on 16S rRNA gene. All the isolates were classified under the iPBE clade, together with six species, i.e., *B. caribensis*, *B. hospita*, *B. phymatum*, *B. sabiae*, *B. steynii and B. terrae*, and uncultured clones identified from largid species in previous studies [31,33]. The congruence of the result with that of the culture-independent approach [31] confirmed that the five isolates were certainly gut symbiotic *Burkholderia* of the bordered plant bug *P. gutta*.

### 3.3. Complete Genomes of the Isolated Symbionts

The two isolates, PGU16 and PGU19, between which the sequence identity of their 16S rRNA gene sequences was 98.7%, were then subjected to whole-genome sequencing with the PacBio RS II sequencer. The reasons why these two isolates were chosen are as follows: PGU16 is the first isolate from *P. gutta*; among the four isolates obtained in 2019, PGU19 shows the lowest 16S rRNA gene sequence identity against PGU16. De novo assembly of generated long-sequence-reads resulted in four circular and one non-circular replicons for PGU16 and five circular ones for PGU19. After closing a gap in the replicon of PGU16 by Sanger-sequencing, we obtained two complete genomes of the *Burkholderia* symbionts.

Based on the synteny between the PGU strains and their closely related species *B. caribensis*, which have two chromosomes and two plasmids, we assigned the five replicons to chromosomes or plasmids: the longest three replicons of each PGU strain corresponded to chromosomes 1 and 2, and plasmid 1 of *B. caribensis*, respectively; the remaining two replicons, which did not correspond to plasmid 2 of *B. caribensis*, were assigned as plasmids.

Both genomes consist of two circular chromosomes and three circular plasmids. The genome of PGU16 and PGU19 are 9,470,940 bp (141 × genome coverage) and 11,246,397 bp (86 × genome coverage) long and contain 8498 and 10,280 protein-coding genes, respectively (Figure 3A,B). Since the genome size of *B. insecticola* RPE64 isolated from the bean bug is 6.96 Mb [15,35], the iPBE symbionts have 1.36- and 1.62-times larger genome than RPE64. The statistics of these genomes and those of the SBE *Burkholderia* symbionts of other stink bugs are summarized in Table 2. The comparison of COG functional categories between the new isolates and other *Burkholderia* is shown in Figure 3C.

The genome-based molecular phylogeny shown in Figure 4 supported that the iPBE clade is a monophyletic subclade within the PBE clade and includes the two isolates from the bordered plant bug. The dDDH value between the isolates PGU16 and PGU19 was 47.7%. The highest dDDH value between the isolates and the described iPBE Burkholderia species with complete genomes was 57.7% (PGU19 vs. *B. terrae*). These values are below the proposed criterion [67], indicating that these isolates are of different species and different from any other described species of the *Burkholderia* iPBE clade.

### 3.4. Virulence and Symbiosis Factors Conserved in the iPBE and SBE Burkholderia Symbionts

To characterize the genomes of the stink bug symbionts that belong to the iPBE clade and/or the SBE clade of *Burkholderia*, comparative genomic analysis was performed. The seven complete genomes, including the five symbionts of stink bugs listed in Table 2, were compared and their orthologous genes were clustered into orthogroups. The other two genomes were *B. pseudomallei* K96243 and *B. phytofirmans* PsJN which were chosen as well-characterized representatives of BCC&P and PBE clades, respectively. In total, 53,870 protein sequences from seven genomes were clustered into 12,556 orthogroups that includes 3795 singleton genes without any orthologous gene in the analyzed data set.

We first searched whether the five symbiont genomes carry the representative virulence or symbiosis factors previously reported in pathogenic species of BCC&P, such as *B. pseudomallei*; plant growth-promoting rhizobacteria (PGPR) of PBE, such as *B. phytofirmans*; and bean bug-associated SBE, such as *B. insecticola* [17,53,54,55,56,57,58,59,60,61,62,63,64,65,66] (Table 3).

The five symbionts consistently possessed the following factors that are commonly important in both pathogenesis and symbiosis: flagella, chemotaxis, lipopolysaccharide (LPS) biosynthetic cluster, phospholipase C, MucD Ser protease, 1-aminocyclopropane-1-carboxylic acid (ACC) deaminase, and all the symbiosis factors reported in *B. insecticola*. Type six secretion system (T6SS), type one fimbriae, and tight adherence (Tad) pili were also commonly found in all the five symbionts; however, the symbiont genomes possessed clusters of more than one for these factors and they did not show one-to-one orthologous correspondence. On the other hand, the capsular polysaccharide synthesis and export cluster and 3-hydroxypalmitate methyl ester (3-OH-PAME)-based quorum sensing system were missing in the five symbionts. Notably, although the two iPBE symbionts possessed type three secretion system (T3SS) and *N*-acyl homoserine lactone (AHL)-based quorum sensing system, the three SBE symbionts did not. The genes related to nodulation and nitrogen fixation, *nodA* and *nifH*, of *B. phymatum* STM 815 were not found in PGU16, PGU19, and the SBE symbionts.

### 3.5. Identification of the Genes Specifically Conserved in the iPBE and SBE Burkholderia Symbionts

To distinguish between the specific genes conserved among the iPBE and SBE symbionts, we then classified the 12,556 orthogroups according to their presence in genomes. The analysis showed that the seven *Burkholderia* symbionts shared 2799 orthogroups and that 282 orthogroups were conserved in the five insect symbionts but were absent in the other two species (Figure 5A). The 282 orthogroups conserved in the five insect symbionts are listed in Appendix A. The number of orthogroups conserved in only the iPBE and SBE symbionts were 603 and 338, respectively (Figure 5A).

Figure 5B shows the comparison of COG functional categories between the above four classes of orthogroups, i.e., conserved in the seven species, conserved in only the five insect symbionts, conserved in only the two iPBE symbionts, and conserved in only the three SBE symbionts. Almost half of the orthogroups conserved in only the iPBE and SBE symbionts were functionally unknown. Figure 5C shows the distribution of the genes classified into the four classes on the genome of the iPBE symbiont PGU16. The chromosome mainly harbors genes conserved in all the seven genomes, as expected. On the other hand, most of the genes conserved in only the five symbionts and the iPBE symbionts were located on the plasmids. Notably, on plasmid 1, the region that harbors the genes conserved in the five insect symbionts was found. The conserved gene region on plasmid 1 consisted of about 40 genes (PPGU16_69460–PPGU16_69920) and interestingly, this 40-gene cluster was also conserved on the genomes of the other four insect symbionts (Appendix A), which is similar to the “symbiosis island” reported in rhizobia [68,69]. This gene cluster contained coenzyme pyrroloquinoline-quinone (PQQ) synthase and ribulose 1,5-bisphosphate carboxylase/oxygenase (Rubisco) (Appendix A). Moreover, a partial duplicate of this conserved region was found only on plasmid 2 of the iPBE symbiont PGU19 (Appendix A).

## 4. Discussion

In this study, we first isolated *Burkholderia* symbionts from the largid stink bug, *P. gutta*, which belong to the iPBE clade of the genus *Burkholderia*. We subsequently determined the complete genomes of two isolated symbionts of *P. gutta* and then compared them with the whole genomes of other three SBE symbionts. Studies on an SBE symbiont, *B. insecticola* RPE64, of the bean bug have revealed that several symbiosis factors, such as genes related to motility and cell wall synthesis, are important for colonization in the midgut crypts [17,57,58,59,60,61,62,63,64,65,66]. The presence of all these symbiosis factors was confirmed in five stink bug-associated *Burkholderia* symbionts, which include two iPBE symbionts (Table 3). This result suggests that: (i) the gut luminal environments are fundamentally similar between the largid bug and the bean bug; and thus (ii) the iPBE symbionts can colonize the gut of largid bug upon the similar mechanism reported in *B. insecticola* [11,20]. Our recent comparative transcriptome analysis between the cultured and the gut-colonizing *B. insecticola* suggests that the symbiont plays a role in recycling host’s nitrogen wastes, such as allantoin and urea, and providing the host bean bug with essential nutrients [58]. Considering that the iPBE symbionts carry the pathways for allantoin and urea metabolisms, nitrogen recycling may be a metabolic function of the symbionts in largid bugs. It should be noted that the previously-reported symbiosis factors were also conserved in the non-symbiont *Burkholderia* species, *B. pseudomallei* and *B. phytofirmans* (Table 3). Considering that chinch bugs, *Blissus insularis* and *Cavelerius saccharivorus* (superfamily Lygaeoidea; family Blissidae), are associated with all BCC&P, PBE, and SBE *Burkholderia* [25,70] and that the genus *Pandoraea*, the sister group of *Burkholderia* sensu lato, can be beneficially associated with the bean bug under the laboratory condition [18], key symbiosis factors for association with stink bugs might be originated at the common ancestor of *Burkholderia* sensu lato or earlier.

Our comparative analysis also identified 282 orthogroups that were specifically conserved in five insect symbionts (Figure 5A and Appendix A). These specifically conserved genes, of which more than one third are functionally unknown (Figure 5B), may include important genes that are essential for environmental adaptation inside insect guts, although the 282 orthogroups do not contain any of the symbiosis factors reported in the bean bug symbiosis to date. Interestingly, a gene mapping analysis revealed a conserved gene region that consists of about 40 genes in a plasmid and a second chromosome of the symbionts (Figure 5C and Appendix A). This conserved region, despite its relatively small size, is reminiscent of the “symbiosis island” or “symbiotic plasmid” of the nodule-forming rhizobia symbionts in leguminous plants [68,69]. Although we have identified only 527 genes of *B. insecticola* RPE64 (~8% in the whole genome) that are upregulated in the midgut crypts of the bean bug compared to those cultured conditions [58], more than half of the genes of this “symbiont-conserved gene cluster” were among these genes (Appendix A). Furthermore, the conserved region contains functionally interesting genes: PQQ-dependent dehydrogenase and a PQQ biosynthesis cluster that consists of four genes. In *Drosophila*, the PQQ-dependent alcohol dehydrogenase activity of a commensal gut bacterium, *Acetobacter pomorum*, has been experimentally demonstrated; it modulates insulin/insulin-like growth factor signaling in the host to regulate metabolic homeostasis of the insect host. Moreover, gene-deletion mutants of the PQQ biosynthesis and PQQ-dependent alcohol dehydrogenase caused retarded growth and smaller body size [71]. Considering the genomic conservation in the five insect symbionts and higher expression under in vivo conditions, it is plausible that these PQQ genes play a similar function in the stink bug-*Burkholderia* symbiosis. It would be of great interest to test this possibility in the bean bug model system and the symbiosis between largid bug and iPBE symbionts.

From the difference between the iPBE and SBE symbionts, it should be noted that T3SS and AHL-based quorum sensing are found only in the iPBE symbionts (Table 3). In the genome of the notorious human pathogen *B. pseudomallei* K96243, three T3SS clusters have been known; one of them plays a role in virulence, whereas the function of other two clusters in virulence has been less characterized [53,72]. Although the biological functions of the two T3SS clusters remain unclear, it is notable that the T3SS cluster found in the iPBE symbionts is orthologous to one of these less characterized T3SS. Although a previous comparative genomic study by Angus et al. [56] reported that PBE *Burkholderia* generally lack T3SS clusters, a recent genomic study revealed that T3SS clusters that are orthologous to the less characterized one are conserved in the iPBE species, *B. caribensis*, *B. terrae*, and *B. hospita* [73]. Apart from *Burkholderia*, T3SS plays a pivotal function in diverse symbiotic systems, such as the legume-*Rhizobium* and weevil-*Sodalis* symbioses [74,75]. Probably, the presence of the T3SS cluster is a genomic signature of the iPBE *Burkholderia* and plays a specific role in the symbiosis. Quorum sensing is also known as an important factor in various symbiotic systems, including the squid-*Vibrio* symbiosis and root nodule symbiosis of leguminous plants [76,77,78]. Therefore, these factors possibly play important roles also in the largid bug-*Burkholderia* gut symbiosis. If we are to set-up an experimental system for the bordered plant bug symbiosis, these factors, i.e., T3SS and quorum sensing, would be the first candidates for functional analyses.

The iPBE clade was highly supported in the genome-based phylogeny (Figure 4) and includes not only the stink bug symbionts isolated in this study but also plant-associated species such as *B. caribensis* and *B. phymatum* [79,80,81]. This result suggests that other members of this clade are also associated with largid bugs. Such a polygamous nature of the iPBE *Burkholderia* might be underpinned by their remarkably large genome. *Burkholderia* species generally possess larger genomes, with an average of 7–8 Mb, than other bacteria [12,82]. Meanwhile, iPBE species possess even larger genomes than other *Burkholderia* species: the genome size of *B. terrae* DSM 17804^T^, *B. hospita* DSM 17164^T^, *B. caribensis* MWAP64^T^, and *B. phymatum* STM815^T^ are 10.1 Mb, 11.5 Mb, 9.0 Mb, and 8.7 Mb, respectively [42,73,83]. This is also true for the PGU strains (Table 2). As mentioned in previous studies, larger genomes of *Burkholderia* may be advantageous for their adaptability [23,84]. At this stage, however, it is unknown whether the plant-associated iPBE species can be also associated with the largid bugs; a comprehensive inoculation test as shown in the bean bug *R. pedestris* should be performed [18]. It should also be tested whether the iPBE symbionts of largid bugs can be a symbiont of plants, although the PGU strains lack *nodA* and *nifH* genes that would have enabled them to induce nodules in leguminous plants.

Due to the advantage that the *Burkholderia* symbiont can be genetically manipulated, the bean bug-*Burkholderia* symbiosis has been widely recognized as one of the promising models for experimentally elucidating host-symbiont molecular interactions [11,12,20]. As several species of the iPBE clade *Burkholderia* can be genetically manipulated [85,86], the genetic manipulation of the iPBE symbionts of largid bug (i.e., the PGU strains) is probably applicable as well. As we start to propagate the bordered plant bug in laboratories, we would be able to experimentally examine the functions of the candidate genes in the largid bug-*Burkholderia* symbiosis in the future. The new symbiotic system, as well as the newly isolated strains with complete genome, would be a useful resource for revealing the complex evolution of insect-microbe symbiosis maintained by environmental symbiont acquisition.

## Figures and Tables

**Figure 1 genes-11-00744-f001:**
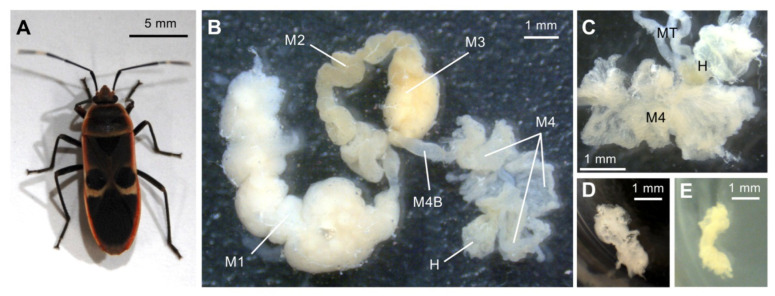
The symbiont-harboring midgut crypts of *Physopelta gutta*. (**A**) An adult female *P. gutta*. (**B**) The dissected midgut of an adult female *P. gutta*. The *Burkholderia* symbiont is specifically localized in the 4th section of the crypt-bearing midgut [31]. (**C**) An enlarged image of the M4 section. (**D**,**E**) Pre-culture of the midgut crypts in yeast-glucose medium. The midgut crypts before pre-culture (**D**) and after two days of pre-culture (**E**). After pre-culture, the crypts became cloudy due to the bacterial growth inside the crypts. Bacterial growth was also observed outside the crypts. M1, midgut 1st section; M2, midgut 2nd section; M3, midgut 3rd section; M4, midgut 4th section; M4B, M4 bulb; MT, malpighian tubule; H, hindgut.

**Figure 2 genes-11-00744-f002:**
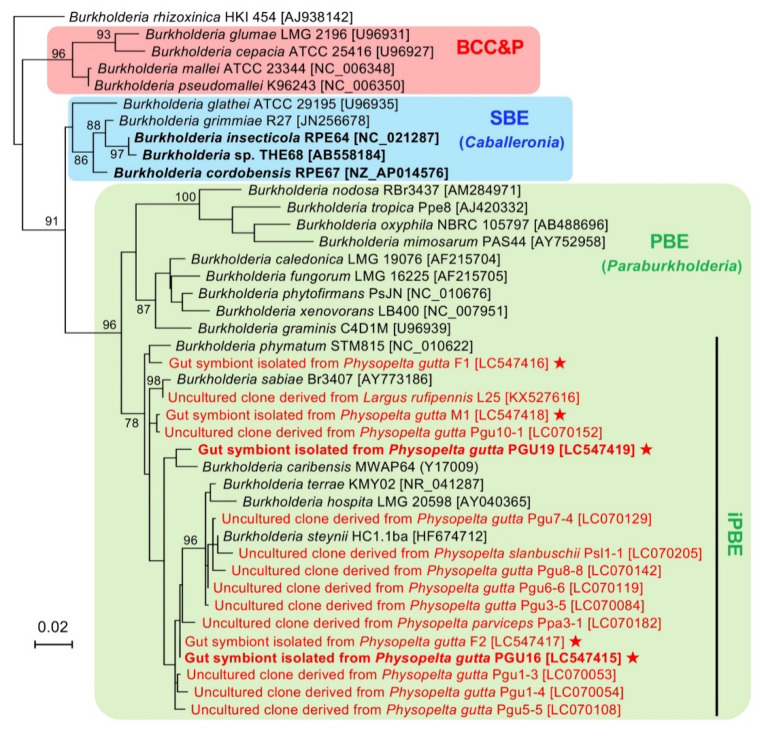
Molecular phylogeny of the *Burkholderia* symbionts isolated from *P. gutta* based on 16S rRNA gene. The tree shows an ML phylogeny of the five isolates and related species/clones of the genus *Burkholderia* sensu lato. The multiple sequence alignment of 1320 nucleotide sites of the bacterial 16S rRNA gene was analyzed. Accession numbers in the DDBJ/EMBL/GenBank DNA database are shown in square brackets. The isolates and uncultured clones of symbiotic *Burkholderia* derived from largid stink bugs are shown in red. Stars indicate the isolates reported in this study. Bold indicates the *Burkholderia* symbionts of stink bugs with complete genome sequences. Bootstrap support values higher than 70% are shown on the internal branches. BCC&P, *Burkholderia cepacia* complex and *Burkholderia pseudomallei* clade; SBE, stink bug-associated beneficial and environmental clade; PBE, plant-associated beneficial and environmental clade; iPBE, insect-associated PBE clade.

**Figure 3 genes-11-00744-f003:**
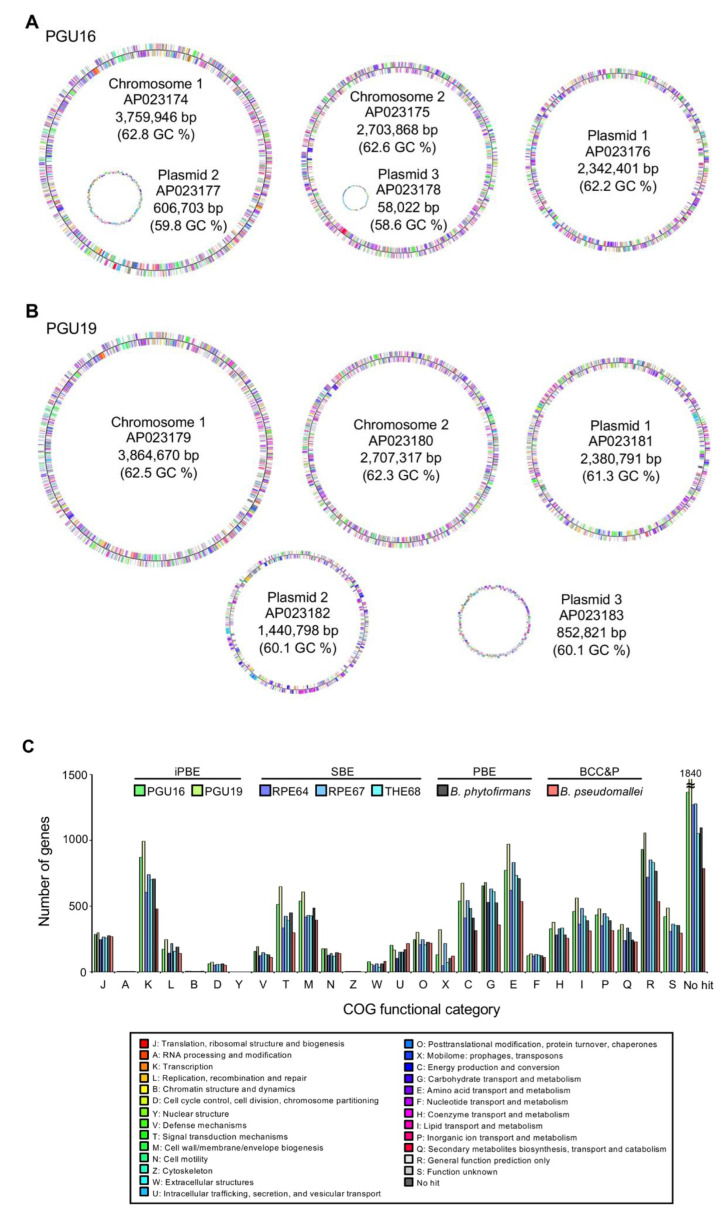
Complete genomes of the isolated iPBE *Burkholderia* symbionts. Circular visualization of the genomes of *Burkholderia* sp. strain PGU16 (**A**) and *Burkholderia* sp. strain PGU19 (**B**). Colors indicate the Clusters of Orthologous Group (COG) functional categories of the genes on the genomes. (**C**) The number of genes classified into each COG functional category in the five symbionts of stink bugs, *Burkholderia phytofirmans* PsJN and *B. pseudomallei* K96243. A gene may be classified into two or more categories; therefore, the sum of numbers here might be different from the total number of genes in a genome. The description of each COG category is shown in the bottom. “No hit” indicates no homologous sequence was detected in the COG database with blastp.

**Figure 4 genes-11-00744-f004:**
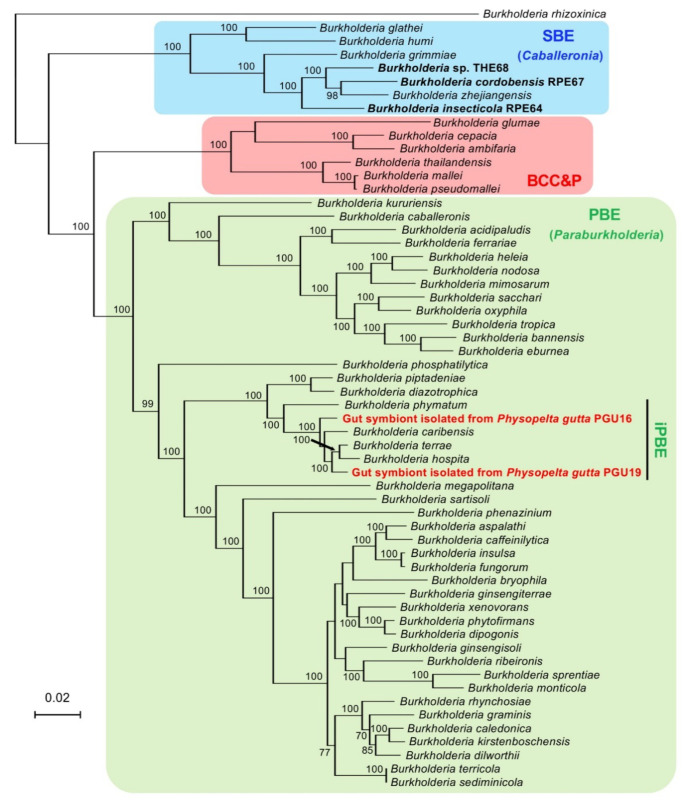
Genome-based molecular phylogeny of the *Burkholderia* symbionts isolated from *P. gutta.* The tree shows an ML phylogeny of the two isolates and related species of the genus *Burkholderia* sensu lato. The multiple sequence alignment of 22,112 amino acids of the 70 UBCGs was analyzed. Accession numbers in the DDBJ/EMBL/GenBank DNA database are shown in Appendix A. Isolates of the symbiotic *Burkholderia* derived from the largid stink bug are shown in red. Bold indicates the *Burkholderia* symbionts of stink bugs. Bootstrap support values higher than 70% are shown on the internal branches. BCC&P, *B. cepacia* complex and *B. pseudomallei* clade; SBE, stink bug-associated beneficial and environmental clade; PBE, plant-associated beneficial and environmental clade; iPBE, insect-associated PBE clade.

**Figure 5 genes-11-00744-f005:**
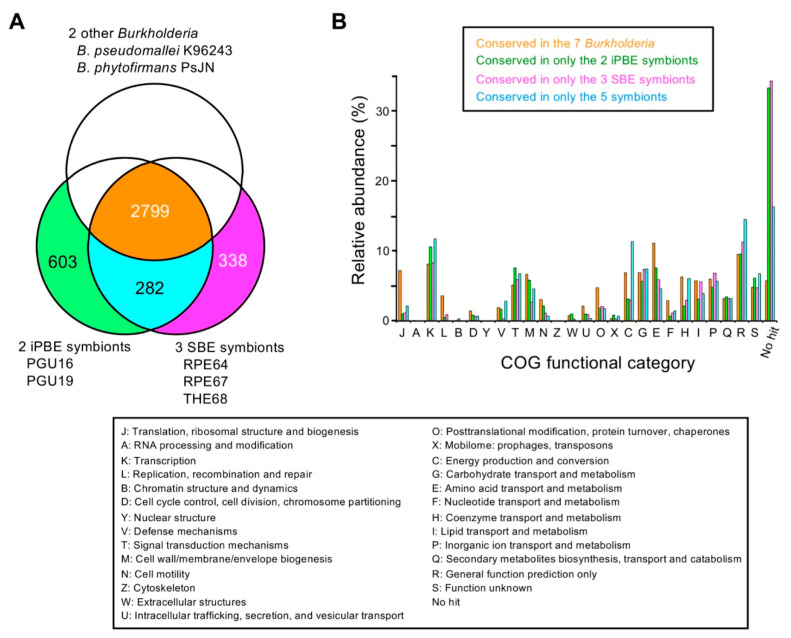
Comparative genomics for highlighting conserved genes among the five *Burkholderia* symbionts of stink bugs. (**A**) Venn diagram for the comparison between the five symbionts and the other two *Burkholderia*. The number of orthogroups is shown. One orthogroup may include two or more genes from a genome. (**B**) Relative abundance of the conserved orthogroups classified into each COG functional category. Since one orthogroup may be classified into two or more categories, the sum of relative abundance can be over 1. The description of each COG category is shown below the panels A and B. (**C**). The distribution of the conserved genes on the genome of the iPBE symbiont PGU16. The arrowhead indicates the region that contains the cluster of conserved genes in the five symbionts.

**Table 1 genes-11-00744-t001:** Insect samples and isolated *Burkholderia* symbionts.

Host Physopelta gutta				Symbiotic Burkholderia		
Sex ^1^	Sampling Date	Sampling Location (in Okinawa, Japan)	Dissection Date	ID	Acc. No. ^2^	The Closest Species (% Identity) ^3^
F	28 June 2016	Sesoko island	1 July 2016	PGU16	LC547415	Burkholderia caribensis MWAP64 (99.3)
F	29 June 2016	Kunigami	1 July 2016	— ^4^	— ^4^	— ^4^
F	4 July 2019	Nago	18 July 2019	F1	LC547416	Burkholderia sabiae Br3407 (99.0)
F	4 July 2019	Nago	18 July 2019	F2	LC547417	B. caribensis MWAP64 (99.3)
F	4 July 2019	Nago	18 July 2019	— ^4^	— ^4^	— ^4^
M	4 July 2019	Nago	18 July 2019	M1	LC547418	B. sabiae Br3407 (99.3)
M	4 July 2019	Nago	18 July 2019	PGU19	LC547419	B. caribensis MWAP64 (99.4)
M	4 July 2019	Nago	18 July 2019	— ^4^	— ^4^	— ^4^

^1^ F, Female; M, Male. ^2^ Accession number of sequences of 16S rRNA gene is shown. ^3^ The result of the EzBioCloud [39] is shown. ^4^
*Burkholderia* symbiont was not isolated.

**Table 2 genes-11-00744-t002:** Statistics of the complete genomes of *Burkholderia* symbionts of stink bugs.

	*Burkholderia* sp. PGU16	*Burkholderia* sp. PGU19	*B. insecticola* RPE64	*Burkholderia cordobensis* RPE67	*Burkholderia* sp. THE68
Clade	iPBE	iPBE	SBE	SBE	SBE
Host species	*Physopelta gutta*	*Physopelta gutta*	*Riptortus pedestris*	*Riptortus pedestris*	*Togo hepipterus*
	(Pyrrhocoroidea: Largidae)	(Pyrrhocoroidea: Largidae)	(Coreoidea: Alydidae)	(Coreoidea: Alydidae)	(Lygaeoidea: Rhyparochromidae)
Genome size (Mb)	9.47	11.25	6.96	8.69	7.98
GC content (%)	62.4	61.7	63.2	63.4	63.1
Chromosome	2	2	3	3	4
Plasmid	3	3	2	3	2
CDSs ^1^	8498	10,280	6349	8156	7278
Coding ratio (%) ^1^	85.0	81.8	87.7	87.8	87.9
Average protein length (aa) ^1^	315.8	298.2	320.5	311.8	321.2
rRNAs ^1^	12	14	10	15	16
tRNAs ^1^	71	70	69	73	68
Reference	this study	this study	[15,35]	[36]	[37]

^1^ Statistics calculated with DFAST [43] are shown.

**Table 3 genes-11-00744-t003:** Presence/absence of the virulence and symbiosis factors in seven *Burkholderia* genomes.

		iPBE		SBE			BCC&P	PBE
Functions ^1^	Factors ^2^	*Burkholderia* sp. PGU16	*Burkholderia* sp. PGU19	*B. insecticola* RPE64	*B. cordobensis* RPE67	*Burkholderia* sp. THE68	*B. pseudomallei* K96243	*B. phytofirmans* PsJN
Flagella	VPS	+	+	+	+	+	+	+
Chemotaxis	VPS	+	+	+	+	+	+	+
T3SS ^3^	VP	+	+	-	-	-	+	-
T6SS ^3^	VP	+	+	+	+	+	+	+
Capsular polysaccharide synthesis and export cluster	V	-	-	-	-	-	+	-
LPS biosynthetic cluster ^3^	V	+	+	+	+	+	+	+
Phospholipase C	V	+	+	+	+	+	+	+
Metalloprotease A	V	-	-	-	-	-	+	-
MucD Ser protease	V	+	+	+	+	+	+	+
Type 1 fimbriae	V	+	+	+	+	+	+	+
Type 4 pili	VP	+	-	+	+	+	+	+
Tad pili ^3^	V	+	+	+	+	+	+	+
T4SS ^3^	VP	+	-	-	-	+	-	+
Siderophore biosynthesis	P	+	-	+	+	+	+	+
ACC deaminase ^3^	P	+	+	+	+	+	+	+
AHL-based quorum sensing ^3^	VP	+	+	-	-	-	+	+
3-OH-PAME-based quorum sensing ^3^	P	-	-	-	-	-	-	-
PHA biosynthesis ^3^	S	+	+	+	+	+	+	+
Purine biosynthesis	S	+	+	+	+	+	+	+
Allantoin and urea metabolic pathway	S	+	+	+	+	+	+	+

^1^ The following symbiotic genes reported in the stink bug-*Burkholderia* symbiosis were conserved in all the *Burkholderia* genomes and were omitted: *uppP*, *amiC*, *wbxA*, *waaF*, *waaC*. ^2^ V, Virulence; P, Plant growth promotion; S, Symbiosis with the bean bug. ^3^ T3SS, Type three secretion system; T6SS, Type six secretion system; LPS, Lipopolysaccharide; Tad, Tight adherence; T4SS, Type four secretion system; ACC, 1-aminocyclopropane-1-carboxylic acid; AHL, *N*-acyl homoserine lactone; and 3-OH-PAME, 3-hydroxypalmitate methyl ester; PHA, Polyhydroxyalkanoates.

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
