# Peer review of "Genomic Comparison of Insect Gut Symbionts from Divergent Burkholderia Subclades"

_genes, 2020, doi:10.3390/genes11070744_

Round 1
Reviewer 1 Report
Summary
Building on the thoroughly characterized symbiosis between specific groups of stinkbugs and Burkholderia bacteria, this manuscript uses genomic information from symbionts of distinct Burkholderia subclades, and from different host families, with the aim of identifying bacterial genes playing a key role in the evolution of symbiosis between these partners. While the study does not conclusively answer this challenging question, it gives important and significant steps to explore this matter. By comparing previously known genomes of symbionts from the stinkbug-associated Burkholderia subclade (now Caballeronia), to newly sequenced genomes of stinkbug symbionts falling in the plant-associated Burkholderia (now Paraburkholderia), as well as free-living members of this subclade, the study reveals that: (i) several formerly identified factors essential for symbiosis are common across symbiotic and non-symbiotic Burkholderia, (ii) a set of 282 genes are exclusively shared between the symbiotic strains yet most are of unknown function and (iii) bordered plant bugs are confirmed to associate with strains that are closest to plant-associated than to other stinkbug-associated Burkholderia. The study puts forward a very interesting and relevant evolutionary question, addressed using genomic tools and in a very suitable system for this purpose. I only have a few comments that might improve the accuracy and coherence of some aspects of the study, as well as some minor suggestions in the text.
General comments
- Given that the function of most of the identified shared genes is unknown and therefore mechanistic connections with the evolution of symbiosis are not clearly built in the manuscript, I suggest to change the title to align it better to the content. E.g. “Genomic comparison of insect gut symbionts from divergent Burkholderia clades and their free-living relatives”, or similar.
- The assignment of the 2nd, 3rd and possibly 4th biggest replicons in both PGU16 and PGU19 as plasmids is rather surprising for several reasons, yet this is not mentioned in any way in the manuscript. Plasmids are usually much smaller in size (to my knowledge the largest circular megaplasmids reported are ~1.5Mb). Also, many Burkholderia are known to have multiple chromosomes, and some of the results in the present manuscript ―like the location of the shared 40-gene cluster― are also suggestive of these being chromosomes. Given that genes located on plasmids are often acquired through recent horizontal gene transfer, and their genomic signatures including GC content and dinucleotide composition are usually different from the chromosome (diCenzo GC, Finan TM. The Divided Bacterial Genome: Structure, Function, and Evolution. Microbiol Mol Biol Rev. 2017;81(3):e00019-17), these characteristics might be useful to make this assignment. Sequencing coverage can (potentially) also be relevant, given that multi-copy plasmids are expected to have higher coverage. In turn, chromosomes are expected to have very similar coverage. As suggested in the specific comments, I recommend to provide this relevant information and also carefully revise and justify the replicon assignments.
- I suggest providing more insight in the abstract about the nature of the genes shared between the symbionts. A couple of sentences mentioning that most of the shared genes have unknown function, and that previously identified symbiotic factors are also shared with other Burkholderia are key findings.
- It would be worth to mention in the discussion the interesting cases of blissid bugs (also Lygaeoidea) like Cavelerius saccharivorus and Blissus insularis, which carry Burkholderia strains from all major subclades (PBE, SBE and BCC&P). This might suggest that key symbiotic factors for association with stinkbugs might be widespread across Burkholderia, or at least a few additional convergent cases are present within Burkholderia sensu lato.
Specific comments
- Line 14: consider removing “…species and”, to refer to strains in general.
- Line 34: I suggest stating as “…one million described species”.
- Line 38: I suggest to remove “the” before “beneficial symbionts”.
- Line 39: It is probably more accurate to refer to “most characterized insect symbionts”
- Line 40: since it is not vertical transmission per se that causes genome reduction, consider replacing “… which thereby have made their genomes reduced” by “which is often coupled with symbiont genome reduction”.
- Lines 40-41: Consider re-phrasing for more precision to “The longer evolutionary history of most of these characterized associations has made it difficult to reveal the evolutionary transition…” or similar.
- Lines 42-43: I agree that revealing evolutionary transitions can be hard in many systems, but I believe that the cases mentioned here are not the only examples for which we have information about the evolutionary origin of symbionts. The authors may consider removing this sentence. Instead, it would be worth already highlighting here that there are few insect systems (like this one) in which both the symbionts and very closely related free-living strains are available as in vitro cultures with corresponding whole-genome sequence data. I believe this underlines better the importance of the study.
- Line 44: a letter is missing, “… symbiotic organS”.
- Line 51: I suggest to write “…symbiontS that ARE later…”
- Line 72: consider changing “would provide a great opportunity in tracking” for “provides a valuable opportunity for tracking”
- Line 86: for clarity, I suggest to replace “remain absent” for “were lacking”.
- Line 96: please include “The” before “Burkholderia symbiont…”
- Line 150: for clarity, I would replace “a non-circular replicon” for “one of the replicons”
- Lines 157-158: this might not be the most reliable way for assigning each replicon as chromosomes or plasmids. Please see the general comments.
- Line 165: I suggest to include “the” before “maximum likelihood method”.
- Lines 176-177: Consider re-organizing this sentence for clarity, first referring to the GTR model for the genome-based phylogeny including 70 core genes, and then the WAG for the 16S rRNA gene phylogeny.
- Lines 215-219: please include information here and/or in the discussion on whether the recovered isolates should be representative based on the previous culture-independent study. That is, how frequently these strains are expected to be found in the insect populations, in case this information is available. If unknown, this is also worth mentioning.
- Lines 231-232: If I understand correctly, this phylogeny is based on one alignment (of multiple sequences). Please refer to a single alignment in this case, i.e “The sequence alignment of 1,320… was analyzed”.
- Lines 246-247: please see the general comments regarding the assignment of replicons as chromosomes or plasmids.
- 3: I strongly suggest to either map or report GC content, as this might provide useful information per replicon. Also, consider including the COG category descriptions directly in the legend/box (not in the caption), since currently there is a lot of blank space in the figure that can be used more informatively and facilitates its navigation.
- Line 271: I would explain in the text or here what is meant by “General function prediction only” in relation to COG category R.
- Line 275: please add “The” before “Genome-based phylogeny…”
- Table 3: please specify what is included in “surface polysaccharide biosynthetic cluster”, specially in differentiation to the LPS immediately above.
- Lines 335-338: this result should be framed differently after validating the plasmid/chromosome assignments.
- Lines 348-349: I suggest including the COG descriptions in the caption again to facilitate reading the figure.
- Lines 353-359: although the cultivation of the symbionts is a valuable achievement, I believe it is not central for the discussion and already explained above. Consider mentioning this in less detail for this section.
- Lines 361-363: I would briefly recapitulate these symbiotic factors to provide sufficient background around this point.
- Lines 363-371: it might be worth to explicitly add here that, since these genes are also present in the free-living Burkholderia (correct?), they might be necessary but not exclusive to the symbiotic strains among the clade.
- Line 383: I suggest making more clear that this refers to the same experiment, i.e. “were among these genes”.
- Line 390: consider mentioning specifically how similar the gene cluster for PQQ biosynthesis and the related dehydrogenase are between the 5 symbionts.
- Lines 415-416: Rephrase as "members of this clade" since otherwise it implies that exactly those species/strains were found in the bugs, which is contradicted below, and also not shown by the results.
- Line 427: “nodules” instead of “a nodule”
Reviewer 2 Report
Takeshita and Kikuchi undertake an interesting and informative comparative genomic study of Burkholderia bacterial symbionts of the stinkbug in order to study the evolutionary transition to this insect symbiont lifestyle and identify candidate genes involved in this symbiosis. In this study the authors isolate and sequence two additional Burkholderia stink bug symbiont genomes. These two genomes are in a different phylogenetic clade/genus of Burkholderia (the Plant Beneficial and Environmental [PBE] clade [also known as the genus Paraburkholderia], with these insect-associated Burkholderia in the group referred to as iPBE) from the previously sequenced three stink bug symbionts (in the Stink Bug associated and Environmental [SBE] clade, also known as the genus Caballeronia). These iPBE symbionts and have relatively large, multi-replicon genomes that are larger than the SBE symbiont genomes. The authors then identified genes unique to these five stink bug symbiont genomes (compared to a pathogenic Burkholderia- (B. psuedomallei) and a plant growth promoting symbiotic Burkholderia (B. phytofirmans). Many of these unique genes are located together on one of the plasmids, likely representing a putative “symbiotic island”. While many of these genes are uncharacterized, some of the iPBE symbiont specific genes are similar to those characterized to be important in the SBE-stink bug symbiosis, suggesting similar functions. This stink bug – Burkholderia symbiosis is ecologically and evolutionarily relevant and is a strong model of how bacterial evolve to become host associated symbionts, and this study adds important new information. The manuscript is very well written and concise. There are some aspects of the study that should be clarified, however, before it is suitable for publication.
One issue that should be addressed is a justification of the other Burkholderia genomes used in the comparative analysis. Why were B. pseudomallei and B. phytofirmas specifically chosen? Was it because they are distantly related to the insect symbionts, well-characterized representatives of pathogenic and symbiotic lifestyles? This is what the analyses would suggest, but a justification/explanation up front would be useful. Including a bit more information in the introduction about these different Burk lifestyles, and the genes involved in them, would help set up the justification for why the authors are looking for specific “symbiotic factors”. There are many Burkholderia comparative genomic studies, so it would be good to mention and cite some previous findings earlier in the manuscript. Furthermore, there are other Burkholderia genomes that have been sequenced that are more closely-related to the iPBE symbionts. Why were they not used? Perhaps this will be a future study to answer a different question, but that should also be clarified. It seems that if the intention is to study the transition to the stink-bug symbiont lifestyle is the goal, the genome comparisons to the more closely-related B. phymatum and/or B. caribenis may be useful as well. I’m not suggesting that the authors need to do this for their study, just that they justify why they made the comparisons they did. Related to that, the following additional clarifications that would be useful:
- How much is known about the iPBE symbionts and their effects on their hosts? The introduction talks about Burkholderia symbionts having a beneficial effect on their insect host (higher numbers of eggs laid and insecticide resistance). It is not clear to me, however, if this has currently only been studied for the SBE symbionts, or if this is also known for the iPBE symbionts. Since in the discussion the authors are inferring function of the iPBE symbionts from the genome sequences it suggests to me that less is known about the iPBE symbionts, and that would be good to mention as another justification of the study in the introduction.
- The authors only sequenced two of the five isolated genomes, but it was not clear why those two in particular where chosen. Please clarify.
- In the characterization of these new iPBE genomes, the authors find five replicons. They call only the first one a chromosome and the rest are considered plasmids, and it is not clear to me why this is the case, especially since the other symbiont genomes have 3 chromosomes and 2 plasmids. Are there no essential genes (rRNA, tRNAs, etc.) on the four plasmids? That is often used as a criterion to differentiate between chromosomes and plasmids. The potential reasoning given in the methods is that only the first replicon did "correspond to a replicon of B. insecticola”. But these are somewhat evolutionary distant (B. insecticola is in the SBE clade), and wouldn't B. caribensis (very closely related in the iPBE clade) be a better comparison for that? I’m not sure why that was not used as a comparison in this case, as it has two chromosomes (sizes 3.7Mb and 2.9Mb – similar in size to these stink bug symbiont iPBE genomes [Pan et al 2016, Genome Announcements]) and two plasmids.
- In the Results at line 275 the authors refer to the iPBE clade as a “monophyletic subclade within the PBE clade”, and I’m not sure I agree on the monophyly. On the phylogeny this “monophyletic” subclade (highlighted by the black bar in their phylogeny) includes B. phymatum, B. caribensis, B. terrae, and B. hospita. These have not yet been demonstrated to be insect-associated. In the discussion the authors argue that it is “strongly” (line 415) likely that they are, but unless there is an evidence for that yet, then I think “strongly” overstates it. And B. terrae and B. hospita have been pretty clearly demonstrated to be fungal symbionts (most recently in Pratama et al 2020, Genome Biology and Evolution). I do agree with the authors that it is worth testing whether these other species are also associated with insects, but until then I think it would be more accurate to refer to the iPBE as a paraphyletic group (because not all species resulting from that common ancestor associate with insects).
- The authors started by framing this as a system for studying the transition to the symbiont lifestyle, but don't really discuss it again in the discussion, although some of the information they do discuss is pertinent. I think that circling back to this idea a bit more directly at the end would improve the manuscript. For example, do the authors think that because the Burkholderia insect symbionts are environmentally acquired, and must survive in multiple environments that their genomes are so large? This would highlight the important role of symbiont transmission (vertical vs. environmental) in symbiont genome evolution. Does this limit potential co-evolution and mutualist tendencies since host-symbiont fates are not as tightly intertwined? Given their locations on the Burkholderia phylogeny, the SBE and iPBE stink bug symbionts represent two separate/independent transitions to the symbiont lifestyle. What does the fact that have some similar putative symbiont genes imply? Could comparisons to more closely related genomes for each symbiont type lend further insight into the evolutionary transition to the symbiont lifestyle?
There are also some smaller corrections that are listed below.
- Line 41: Might be worth elaborating on Muller's ratchet here, and how vertical transmission contributes genome degredation/size.
- Line 42: This last sentence seems a bit random: why do the authors talk about only fungi specifically when there are many bacterial examples of going from pathogen (or environmental bacteria) to mutualist (e.g. see: Sachs et al 2011 Evolutionary transitions in bacterial symbiosis. Proceedings of the National Academy of Sciences, USA, 108(Suppl 2), 10800–10807). I would make this a broader statement. Also, as worded it appears that the cicadas and planthoppers are talking.
- Line 44: “organ” should be “organs”
- Line 66: The authors say there are six nominated genera, but aren’t there only three?
- Line 74: In the manuscript sometimes the symbiosis is referred to as bug-Burkholderia symbiosis, or stink-Burkholderia symbiosis. Should they all be stink bug-Burkholderia symbiosis?
- Line 77: “Was found ^to be^ more complicated…”
- Line 86: “remain” should be “remains”
- Line 185: B. psuedomallei is misspelled. Also, since B. psuedomallei is used for the comparative genomics, it would be good to include it in one, if not both, phylogenies.
- Line 216: “method” should be “methods”
- Line 298: cite previously reported studies.
